# Anticancer Mechanism of *Astragalus* Polysaccharide and Its Application in Cancer Immunotherapy

**DOI:** 10.3390/ph17050636

**Published:** 2024-05-15

**Authors:** Ziqing He, Xiyu Liu, Simin Qin, Qun Yang, Jintong Na, Zhigang Xue, Liping Zhong

**Affiliations:** 1State Key Laboratory of Targeting Oncology, National Center for International Research of Bio-Targeting Theranostics, Guangxi Key Laboratory of Bio-Targeting Theranostics, Collaborative Innovation Center for Targeting Tumor Diagnosis and Therapy, Guangxi Medical University, Nanning 530021, China; paigehoe@163.com (Z.H.); liuxiyu0509@sr.gxmu.edu.cn (X.L.); qinnsimin@sr.gxmu.edu.cn (S.Q.); 202321629@sr.gxmu.edu.cn (Q.Y.); najintong@sr.gxmu.edu.cn (J.N.); 2School of Pharmacy, Guangxi Medical University, Nanning 530021, China

**Keywords:** *Astragalus* polysaccharides, tumor immunotherapy, cancer, immune checkpoint inhibitors

## Abstract

*Astragalus* polysaccharide (APS) derived from *A. membranaceus* plays a crucial role in traditional Chinese medicine. These polysaccharides have shown antitumor effects and are considered safe. Thus, they have become increasingly important in cancer immunotherapy. APS can limit the spread of cancer by influencing immune cells, promoting cell death, triggering cancer cell autophagy, and impacting the tumor microenvironment. When used in combination with other therapies, APS can enhance treatment outcomes and reduce toxicity and side effects. APS combined with immune checkpoint inhibitors, relay cellular immunotherapy, and cancer vaccines have broadened the application of cancer immunotherapy and enhanced treatment effectiveness. By summarizing the research on APS in cancer immunotherapy over the past two decades, this review elaborates on the anticancer mechanism of APS and its use in cancer immunotherapy and clinical trials. Considering the multiple roles of APS, this review emphasizes the importance of using APS as an adjunct to cancer immunotherapy and compares other polysaccharides with APS. This discussion provides insights into the specific mechanism of action of APS, reveals the molecular targets of APS for developing effective clinical strategies, and highlights the wide application of APS in clinical cancer therapy in the future.

## 1. Introduction

Cancer is a significant global health issue owing to its high incidence and mortality rates, imposing a considerable burden on public health systems. As reported in the 2023 data of the World Health Organization, cancer stands as the primary or secondary contributor to death in individuals under 70 years of age in 183 countries [1]. Typical methods of cancer treatment include chemotherapy, radiation therapy, and surgery. Although surgery and radiation play essential roles in treating early-stage cancers, they often fail to meet the needs of patients with more advanced cancer stages. The main reasons for this lack of effectiveness are tumor recurrence and spread. Despite being tailored to a specific type, stage, and location of cancer, various restrictions exist [2]. Surgical resection combined with systemic chemotherapy is the cornerstone of localized cancer therapy. The cytotoxic effects of chemotherapy, the primary component of combination treatment, not only affect cancerous cells but also harm healthy cells, leading to drug resistance and adverse reactions [3,4,5,6]. Extended chemotherapy regimens may result in drug resistance, causing cancer cells to become tolerant to chemotherapeutic drugs, necessitating higher doses for therapeutic effectiveness, a situation that may be exacerbated by increased toxicity [6]. Given these complexities, the discovery of complementary therapies that enhance the efficacy of traditional treatment while reducing adverse effects is crucial. Recent studies have explored the anticancer and anti-inflammatory qualities of conventional plant extracts [7,8]. These extracts have shown potential to enhance the effectiveness of anticancer vaccines and, when used in conjunction with standard chemotherapy and radiation therapy, can improve treatment outcomes and minimize harmful effects through various mechanisms [9,10,11,12].

*Astragalus membranaceus*, commonly known as Huangqi (Chinese name) or Radix Astragali, is a perennial herb of the legume family that is widely distributed throughout temperate regions worldwide. The dried root of Radix Astragali has a long history in China and has been widely used in the treatment of many diseases since the Ming Dynasty [13]. The dried roots of *Astragalus membranaceus* contain complex bioactive constituents, including polysaccharides, flavonoids, *Astragalus* methyl glycosides, amino acids, trace elements, and more than 100 compounds [14]. Recent studies revealed the pharmacological functions of these components, particularly their polysaccharide fractions [15]. *Astragalus* polysaccharide (APS) has been identified as one of the main active ingredients. It consists of nine monosaccharides, including glucose, galactose, arabinose, and rhamnose, and may contain glucuronic acid and galacturonic acid [16]. Due to its water solubility and low-toxicity characteristics, APS has demonstrated significant biological activities in immune activation and antioxidant, antitumor, antidiabetic, anti-inflammatory, and anti-aging properties; thus, it has great potential for drug discovery in the pharmaceutical and nutraceutical fields [17,18,19,20,21,22,23]. Research indicates that APS is safe and effective for cancer treatment because it enhances the body’s immune response, mitigates the side effects of chemotherapy, and activates immune defenses [24,25].

This review focuses on the potential adverse effects of APS in cancer immunotherapy and its interactions with various immunotherapeutic methods, summarizing the findings of studies involving different cancer cell lines, animal models, and human trials. We highlight the ability of APS to promote the growth and activation of immune cells, as well as its role in inducing apoptosis in cancer cells. Additionally, we examine the supportive role of APS in immunotherapy and its synergistic outcomes when combined with immune checkpoint inhibitors (ICIs), overdose immunotherapy, and cancer vaccines to suggest novel approaches and directions for future cancer treatment strategies.

## 2. Anticancer Mechanism of *Astragalus* Polysaccharide

APS has recently attracted considerable attention in cancer therapy, showing great potential for fighting cancer through its unique immune-activating and immune-modulating functions. APS functions through several mechanisms, including the stimulation of immune cells, cell cycle disruption, the inhibition of signal transduction, the activation of cellular autophagy, the inhibition of lipid metabolism, and improvements in the ability of immune cells to adapt to the tumor microenvironment (TME). Together, these mechanisms make APS a promising adjuvant for cancer immunotherapy.

### 2.1. Activation and Regulation of Immune Cells

The influence of APS on immune cell activity in the TME is summarized in Figure 1 and Table 1.

Dendritic cells (DCs) present antigens, trigger immune responses and modulate immunity [44]. These cells transition from an immature state with low major histocompatibility class (MHC) II expression to maturity as they process antigens and move to lymphoid tissues, thereby elevating MHC II levels. Mature DCs display processed antigens in the lymph nodes, present endogenous antigens via MHC I molecules, and process antigens through various pathways, leading to T-cell activation [45,46]. Studies indicate that APS injections induce a significant elevation of MHC-I and MHC-II co-stimulatory molecules in splenic DC subgroups in C57BL/6 mice [26]. APS enhanced DC maturation by stimulating the secretion of interleukin-12 (IL-12) [27] and amplifying the surface expression of CD40, CD80, and HLA-DR in these cells [47]. APS intensified the immune response by enhancing the pDC pathway [28]. APS-loaded nanoparticles have shown the potential for radiotherapy to activate antigen presenting cells (APCs) and reverse TME [29].

When APCs take up and process tumor-specific antigens before migrating to lymphoid organs, they collectively produce both activating and inhibitory signals that trigger the activation of T cells, resulting in the formation of effector T cells targeting tumor-specific antigens. This is a defense mechanism against cancer cell evasion of the immune system [48,49]. APS facilitates T-cell activation [50,51]. In a study by Bamodu et al., APS was shown to boost the transformation of splenic DCs into CD11c (high) and CD45RB (low) DCs, facilitate the shift from a Th2 to Th1 response, and augment the Th1/Th2 ratio, thereby enhancing the immune function of T lymphocytes [30]. Another study by Hwang et al. illustrated that the intranasal administration of APS can activate DCs and further stimulate natural killer (NK) cells and T cells within the lymph nodes [31]. In vivo experiments have shown that APS can enhance immunity in cyclophosphamide-induced immunosuppressed mice by promoting the proliferation of T lymphocytes and B lymphocytes and inducing the production of immunoglobulins (IgA, IgG, and IgM) and a variety of cytokines, including IL-6, IL-2, interferon-gamma (IFN-γ), complement 3, complement 4, and tumor necrosis factor-alpha (TNF-α) [52].

Helper T cells (Th cells) play a crucial role as mediators of the immune response, and are predominantly categorized into Th1 and Th2 cells, which connect and activate other immune cells within the peripheral effector memory T (TEM) cell compartment [53]. The combination of APS and polysaccharide peptides led to expansion of the Th cell subset, a reduction in the proportion of suppressor T cells/cytotoxic T cells, and an increase in the CD4^+^/CD8^+^ ratio. Through the regulation of immune responses and reinforcement of antitumor effects, these interventions influence the levels of Th1, Th17, and Th2 cytokines, augmenting IL-2, tumor TNF-α, IFN-γ, and IL-17A, while diminishing IL-10 levels [32].

Studies have been carried out using nanoparticles as carriers to improve immunotherapy through APS activation of T cells. These specialized nanoparticles target lymph nodes, assist in escaping lysosomes, activate distant DCs, and control T-cell differentiation through cross-presentation [33]. They also trigger specific IgG expression, eliciting a Th1-polarized immune response [34] that effectively combats cancer cell growth, spread, and infiltration [35], ultimately disrupting the immunosuppressive microenvironment and establishing an enduring and strong tumor antigen-specific immunity.

Macrophages possess anti-infective, antitumor, and immunomodulatory properties [54]. Acting as a potent regulator, APS primarily operates through the NOTCH signaling pathway and is closely linked to factors such as IL-6, TNF-α, inducible nitric oxide synthase, and CXCL10 [36]. APS enhanced mucosal bacterial defense in vivo and in vitro by activating Toll-like receptor 4 (TLR-4) and nuclear factor κB (NF-κB)/Rel complexes, strengthening innate immunity during infections. This activation led to macrophage activation and the release of nitric oxide (NO) and TNF-α, which aid in inhibiting cancer cell proliferation [37]. An aqueous extract from *Astragalus* demonstrated protection against methotrexate-induced inhibition of RAW264.7 macrophage proliferation and induced the expression of pro-inflammatory factors associated with RAW264.7 cells [38]. APS shifted the pro-inflammatory and anti-inflammatory signaling balance in favor of an anticancer phenotype by proportionally increasing the M1/M2 macrophage ratio [30]. Evaluation of the anticancer effects of APS in a tissue-engineered tumor model showed that APS-stimulated RAW246.7 macrophages and their supernatants prompted apoptosis and reduced carcinoid tumor size [39].

APS activates macrophages to stimulate B cells, thereby enhancing humoral immune responses [40]. In a study on C4H/HeJ mouse splenocytes, APS effectively induced B-cell proliferation through a pathway that was not TLR3-dependent. This stimulatory effect was primarily mediated by immunoglobulins on the cell surface, offering new insights into the specific mechanisms by which APS regulates B cells within the immune system [18].

As the first line of antitumor immunity, NK cells are vital to the innate immune system. APS regulates the activation and quantity of NK cells, boosting the growth-inhibitory activity of ICIs and immune cell cytotoxicity against tumors. This effect was demonstrated in a mouse model of melanoma. The APS-treated group showed a significant increase in the number of CD3-NK1.1^+^ NK cells within the mesenteric lymph nodes. Additionally, there was a rise in the expression of activation markers by NK cells, such as IFN-γ, granzyme B, and perforin [31,55]. Moreover, APS synergized with tanshinone to enhance the effectiveness of the chemotherapeutic agent carboplatin in inhibiting melanoma cell proliferation. This was accomplished by enhancing the body’s immune response, increasing the proportion of CD4^+^ and CD8^+^ cells in the spleen, and boosting the activity of NK cells and cytotoxic T cells [41].

In the H22 mouse hepatocellular carcinoma (HCC) model, the effect of APS was significant. It triggered apoptosis and reduced further damage to tumor cells while enhancing the levels of cytokines like TNF-α, IL-2, and immune cells, including macrophages, lymphocytes, and NK cells [42]. MHC I chain-associated molecule A (MICA) and B (MICB) are the primary ligands for the activation receptor NKG2D on NK cells. The binding of MICA/B and NKG2D led to NK cell activation by increasing the levels of phosphorylated extracellular signal-regulated kinases. APS also upregulated the expression of MICA and MICB on the surface of H22 tumor cells, stimulating NK cell activation with the release of IFN-γ, granzyme-b, and perforin to promote cancer cell clearance [43].

Traditional cancer treatment methods, such as specific chemotherapy medications, can induce a stress response in malignant cells. The combination of therapeutic approaches, including APS, enhances the expression of NK cell-stimulating proteins in tumor cells, thereby increasing their susceptibility to NK cell-triggered breakdown.

### 2.2. Anti-Proliferative and Apoptosis-Inducing Effects on Cancer Cells

The anti-proliferation and apoptosis-inducing effects of APS on cancer cells are summarized in Figure 2 and Table 2.

#### 2.2.1. APS Interferes with the Cell Cycle

APS can effectively inhibit cancer cell proliferation by blocking the S, G2/M, and G0/G1 phases of the cell cycle. 

APS effectively hindered the growth of cancer cells by blocking their progression through the S phase of the cell cycle. A novel variant of APS, APS4, induced apoptosis in poorly differentiated sarcoma S180 cells and human gastric cancer MGC-803 cells by inhibiting the S phase, and its efficacy increased in a dose-dependent manner [56,57]. Furthermore, nanoparticles carrying APS (APS-SeNPs) significantly increased the number of HepG2 cells arrested in the S phase, resulting in apoptosis via the disruption of mitochondrial function [58].

Apoptosis is triggered through three main pathways: the intrinsic mitochondrial, intrinsic endoplasmic reticulum, and extrinsic death receptor pathways [57]. APS affected the cell cycle and induced apoptosis in cancer cells via the mitochondrial apoptotic pathway [85,86]. Despite exhibiting minimal cytotoxic effects in breast cancer 4T1 cells, APS effectively stimulated macrophage activation [59,61]. Activated macrophages enhanced mitochondrial membrane permeability, increased the Bax/Bcl-2 ratio, and facilitated the release of cytochrome C, thereby amplifying the activities of caspases 3 and 9, which initiate apoptosis. This process also arrested the cell cycle in the G2 phase, effectively inhibiting 4T1 cell growth [59]. Additionally, APS inhibited the proliferation of SW620 colon cancer cells by dominating G2/M phase arrest. APS also protected RAW 264.7 cells from the damaging effects of paclitaxel by altering the drug’s blockage effect in the G2/M phase and impacting the rate of apoptosis [60].

In studies of the G0/G1 phase, APS induced SCG-1 cells to arrest in the G0 phase and MCF-7 cells to arrest in the G1 phase, thus inhibiting the proliferation of cancer cells [61,62]. A study on APS combined with chemotherapeutic anticancer drugs found that the combination of APS and cisplatin intensified the inhibition of the G0/G1- and S-phase cell cycle of the pharyngeal carcinoma cells CNE-1. This combination also increased the expression of p53 [63]. The p53 protein, which is essential for cell cycle checkpoints, especially DNA repair and replication, plays a significant role in various malignant tumors [87].

#### 2.2.2. APS Inhibits Signal Transduction Pathways

APS affects various signaling pathways such as PI3K/Akt, ERK/MAPK, and NF-κB [64,88].

Abnormal activation of the PI3K/Akt pathway in cancer cells enhanced cell survival by phosphorylating downstream target proteins and inhibiting apoptosis [89,90]. When combined with apatinib, APS significantly decreased phosphorylated Akt and MMP-9 expression in AGS human gastric cancer cells, strengthening the antitumor effects of apatinib by inducing autophagy and apoptosis [64]. When paired with Palmarosa sempervirens or radix trichosanthis, APS induces apoptosis in cancer cells, such as B16F10 and A375 melanoma cells, by suppressing PI3K, Akt, and Bcl-2 mRNA expression levels [65,66].

Improper activation of the ERK/MAPK pathway in various cancers is correlated with cell proliferation owing to ERK overphosphorylation [91]. APS enhanced the anticancer properties of apatinib by inhibiting the phosphorylation of RAS, a critical effector of the ERK/MAPK pathway [92]. APS activated the MAPK pathway in mouse macrophages, involving ERK, p38, and JNK, and regulated cell functions such as growth, differentiation, and cell death [67]. In addition, APS enhanced the pro-apoptotic effects of adriamycin in gastric cancer cells by increasing the expression of cysteinyl asparaginase-3, activating the MAPK signaling pathway, inducing DNA fragmentation, and increasing the expression of tumor suppressor genes (SEMA21F, P21WAF1/CIP1, and FBXW7). These findings laid the foundation for further research and development of APS as a chemotherapeutic sensitizer against anticancer drugs [68].

Additionally, APS opposed the activation of NF-κB and decreased the expression of anti-apoptotic genes such as Bcl-2, Bcl-xL, and IAP family members, enhancing cancer cell sensitivity to chemotherapy [69,70]. It effectively reduced mRNA and protein levels of p65 and p50, suppressed the NF-κB pathway’s transcriptional activity, decreased Bcl-xL protein abundance, and altered Cyclin D1 expression in non-small-cell lung cancer (NSCLC) cells. These actions inhibited A549 and NCI-H358 cell proliferation in laboratory experiments and slowed A549 xenograft growth in animal models [69]. In Fang et al.’s research, APS was found to regulate the TLR4/NF-κB pathway by activating the TLR4-TIRAP/Mal-MyD88 pathway and suppressing NF-κB p65 activity by lowering TLR-4, Myd88, Cyclin D, and Bcl-2 protein levels and preventing p65 nuclear translocation in PANC-1 cells. This led to apoptosis induction in PANC-1 cells and inhibited pancreatic cancer progression [70]. Moreover, APS reversed MAPK/NF-κB pathway phosphorylation in bone marrow mesenchymal stromal cells (BMSCs) co-cultured with lung cancer A549 cells, attenuated A549-induced tumorigenesis, and reduced RAS/ERK signaling in the BMSCs by decreasing RAS and ERK protein levels, thereby restraining BMSC proliferation and abnormal morphological changes [71].

The interaction of APS and Salvia extract influenced the TGF-β/MAPK/Smad pathway by modulating microRNA levels, predominantly through the upregulation of miR-145 and the downregulation of miR-21. This process consequently impacted the TGF-β/MAPK/Smad pathway, converting pSmad3L to pSmad3C and increasing pSmad3C protein expression, thereby demonstrating anti-HCC properties [72]. In addition to the classical TGF-β-Smad pathway, TGF-β can also regulate Smad signaling via non-Smad pathways [93]. ERK, JNK, and p38-MAPK can phosphorylate R-Smad (including Smad1/2/3/5/8) and Co-Smad-Smad4 mainly by targeting their linker regions [94]. APS can inhibit hepatocarcinogenesis by modulating TGF-β/Smad signaling through Smad3 phosphorylation mechanisms [73].

#### 2.2.3. APS Induces Apoptosis in Cancer Cells

APS can effectively induce apoptosis in cancer cells by regulating Bcl-2 protein, microRNA, p53 protein, and the Fas receptor [95].

Two typical proteins of the Bcl-2 family, Bcl-2 and Bax, play key roles in cysteine-dependent apoptosis. Bcl-2 is an anti-apoptotic protein located in the nucleus and mitochondrial membrane, and Bax is a pro-apoptotic protein located in the cytoplasm. Bcl-2 inhibits Bax-induced apoptosis, so the decrease in the level of Bcl-2/Bax is considered a pro-apoptotic manifestation [96]. APS induced apoptosis in H22 HCC cells and MDA-MB-231 breast cancer cells by upregulating Bax and downregulating Bcl-2 [74,75]. Similarly, APS can accelerate apoptosis in HepG2 HCC cells by decreasing Bcl-2, β-catenin, c-Myc, and Cyclin D1, which may be related to the Wnt/β-catenin signaling pathway [76]. In both nasopharyngeal and ovarian cancer models, APS was combined with cisplatin to reduce the levels of Bcl-2 and elevate Bax, caspase-3, and caspase-9 levels. APS significantly reduced the Bcl-2/Bax ratio due to the activation of chemotherapeutic agent chemosensitivity in combination compared to treatment with cisplatin alone [77,78].

During tumor development, the expression of various miRNAs changes, leading to corresponding changes in the expression of downstream target genes. miRNAs regulate tumor proliferation, autonomous growth signaling, apoptosis, and other processes that affect tumor progression [97]. MiR-27a is a well-known oncogene in a variety of tumor types [98,99,100]. APS reduced the expression of miR-27a, leading to upregulation of the tumor suppressor gene FBXW7, which ultimately inhibited proliferation and induced apoptosis in OV-90 and SKOV-3 cells [79]. MiR-133a is an oncogene in various cancer types [101,102,103]. APS reduced the expression of MiR-237a by upregulating miR-133a, inactivating the JNK pathway, and exerting anticancer effects in human osteosarcoma MG63 cells [80]. Bioassay analysis showed that the upregulation of miR-195-5p was positively correlated with the survival rate of patients with lung cancer and that APS could inhibit the proliferation and migration of NSCLC cells by increasing the expression level of miR-195-5p [81].

As an important oncogene, the normal function of the p53 protein is an effective barrier to cancer [104]. One of the biological functions of p53 is its ability to induce apoptotic cell suicide, which is central to its role as a tumor suppressor [105]. APS significantly upregulated p53, p21, and p16 in the large-cell lung cancer cell line H460, inhibited the expression of Notch1 and Notch3, and ultimately inhibited proliferation and promoted apoptosis [82].

Fas is a transmembrane receptor that induces apoptosis by crosslinking with agonistic antibodies or Fas ligands (FasL). In addition to its pro-apoptotic function, Fas can also activate many non-apoptotic signaling pathways, and activation of these pathways can lead to increased tumorigenicity and metastasis. Thus, the Fas/FasL system plays an important role in tumor development [106]. APS induced apoptosis in CD133^+^/CD44^+^ co-positive colon cancer stem cells in a concentration-dependent manner via the Fas death receptor pathway [83].

Furthermore, APS can trigger ER stress in tumor cells by decreasing the expression of O-GlcNAc transferase (OGT) and increasing the expression of O-GlcNAcase (OGA). Through this mechanism, the APS-upregulated CHOP pathway led to doxorubicin-induced apoptosis in cancer cells [84].

### 2.3. Other Antitumor Effects of APS

#### 2.3.1. APS Activates Cellular Autophagy

Although various tumor types display diverse genetic and phenotypic characteristics, they often exhibit shared metabolic changes, including alterations in autophagy [107,108]. The suppression of autophagy leads to a significant decrease in tumor cell viability. Combining autophagy inhibitors with anticancer medications can diminish chemotherapy resistance in cancer cells and effectively suppress cancer progression [109,110].

APS can regulate autophagic activity in cancer cells. APS influenced proteins such as beclin-1 and LC3 by activating the PI3K/Akt/mTOR pathway, promoting the formation and maturation of autophagosomes, and accelerating the removal of damaged components from cancer cells. APS also affected the intracellular energy balance and protein synthesis, enhancing the role of autophagy in inhibiting cancer cell growth and inducing apoptosis [64,111,112]. Cellular autophagy can be induced by apatinib or APS alone. However, the combination of the two significantly enhanced the antitumor effect in gastric cancer AGS cells by suppressing the PI3K/Akt/mTOR pathway. Additionally, the combination of apatinib, APS, and the autophagy inhibitor 3-methyladenine effectively inhibited cell growth and increased the rate of apoptosis [64]. At the cellular autophagy level, APS increased autophagic activity in cervical cancer HeLa cells and improved chemosensitivity to cisplatin by upregulating beclin-1, promoting the conversion of LC3I to LC3II, and downregulating p62 [113]. APS reduced xanthine oxidase-induced autophagy in small-cell lung cancer A549 cells by downregulating LC3B and beclin-1 and increasing mTOR expression [111]. APS nanoparticles exhibited cytotoxicity against breast cancer MCF-7 cells by inhibiting autophagy, inducing oxidative stress, generating excess reactive oxygen species, causing mitochondrial damage, and activating apoptotic pathways [112]. 

#### 2.3.2. APS Inhibits Lipid Metabolism

Lipid metabolism plays an important role in tumorigenesis, invasion, and metastasis [114,115]. Sterol regulatory element binding proteins (SREBPs) are key transcription factors that regulate the expression of lipid synthesis, uptake genes, and play a central role in lipid metabolism under both physiological and pathological conditions; high upregulation of SREBPs in various cancers promotes tumor growth [116].

Abnormal lipid metabolism is accompanied by overexpression of related enzymes, aberrant transcription of non-coding RNAs, and activation of oncogenic signaling pathways [117,118,119]. It has been shown that overexpression of silent information regulator-1 (SIRT-1) inhibited SREBP-1 expression and nuclear translocation by suppressing lipid metabolism through activation of AMPK phosphorylation [120]. APS significantly reduces triglyceride and cholesterol levels in prostate cancer cells. By promoting miR-138-5p targeting to its 3’UTR region, APS inhibited the expression of SIRT-1 and SREBP-1, promoted signaling in the miR-138-5p/ SIRT-1/SREBP-1 pathway in PCa and DU145 cells, and suppressed tumorigenesis and lipid metabolism [121].

#### 2.3.3. APS Affects the Tumour Microenvironment and Regulates Cytokine and Hormone Levels

The TME is crucial for tumor progression, invasion, and dissemination because of its intricate composition [122,123]. Recent studies have emphasized the significant impact of APS on modifying the TME and influencing tumor cell proliferation, apoptosis, and motility [16,24,56,61,122,124,125,126,127,128,129].

APS interacts with the TME through multiple pathways, including the modulation of immune cell functions. For example, when combined with 5-fluorouracil (5-FU) for colon cancer treatment, APS decreased tumor size by reducing indoleamine-2,3-dioxygenase (IDO1) expression in the TME and enhancing CD8^+^ T-lymphocyte infiltration [124]. In the S180 mouse model, APS altered anaerobic metabolism within the TME by affecting CD3^+^, CD4^+^, CD8^+^ T cells, and CD19^+^ B cells, leading to tumor cell death [56]. Furthermore, APS co-cultured with HeLa cervical cancer cells boosted peripheral blood mononuclear cell proliferation and adjusted the immune response by reducing suppressive cytokine levels like IL-10 and TGF-β, thereby enhancing antitumor efficacy [125]. In a mouse model of lung cancer, polysaccharides isolated by Li et al. demonstrated antitumor effects in mice with 4-methyl-4-nitro-8-nitrosoguanidine-induced gastric cancer by promoting the proliferation of splenic lymphocytes, enhancing the activity of NK cells, and increasing the levels of IgA, IgG, and IgM, as well as the expression of CD1^+^ and CD2^+^/CD1^+^ in the blood [130].

In addition to immune cells, APS also influences cytokines and chemokines in the TME. It regulated cytokine expression, including nitric oxide and TNF-α, restraining MCF-7 breast cancer cell proliferation by inducing G1-phase cell cycle arrest and apoptosis [61]. In an H22 liver carcinoma model, APS effectively suppressed tumor growth, improved overall health in tumor-bearing mice, and enhanced macrophage phagocytosis. These effects were associated with increased cytokine production and inhibition of IL-10, MDR1 mRNA, and P-glycoprotein by APS [16,24]. In the microenvironment of human HCC, APS blocked chemokine matrix-derived factor-1 and its receptor by inhibiting the activation of the chemokine receptor 4 CXCR4/CXCL12 signaling pathway, restoring cytokine imbalance (elevated IFN-γ expression and reduced IL-4 and IL-10 expression), and inhibiting the immunosuppressive T lymphocyte CD4^+^CD25^+^ Treg cell growth, proliferation, and migration [131].

The effect of APS on cytokines within the TME influences cancer cell migration and invasion. In research on lung cancer, APS was found to decrease levels of VEGF and EGFR while enhancing the activity of immune organs, thus inhibiting the progression and spread of tumors [52,126,127]. At the same time, APS was shown to decrease levels of macrophage migration inhibitory factor, reduce the presence of epithelial–mesenchymal transition markers, and gradually diminish the migratory and invasive capabilities of cancer cells in a dose-dependent manner [128,129]. In colon and breast cancer microenvironments, APS inhibited epithelial–mesenchymal transition and metastasis in mice through the WnT5/β-catenin signaling pathway [132,133], inhibited IDO1 expression, and increased intratumoral CD8^+^ T-cell infiltration [124]. APS enhanced the chemosensitivity of HCC cell lines and inhibited cell migration and invasion in HCC by inhibiting MDR1 and P-glycoprotein efflux pump functions [134].

The effect of APS on the TME involves various pathways that influence cytokines and hormones [55]. In addition to altering the function and activity of immune cells, APS significantly reduces levels of IL-10 and TGF-β while increasing levels of TNF-α, IL-1α, IL-2, IL-6, and IL-12. This regulation affects growth factors like VEGF and EGFR, rejuvenating an immune-suppressed TME.

## 3. APS-Assisted Cancer Immunotherapy

The assisted effects of APS on cancer immunotherapy are summarized in Figure 3.

### 3.1. APS Boosts the Effectiveness of Immune Checkpoint Inhibitors in Therapy

ICIs and natural polysaccharide-based substances have shown promise for overcoming the immunosuppressive environment of tumors and enhancing the efficacy of cancer immunotherapy, particularly against solid tumors. APS has demonstrated potential as a therapeutic agent in cancer treatment by influencing PD-1/PD-L1 signaling and related pathways [55,66,135,136,137]. The specific mechanism was demonstrated by the fact that S12, a single-chain variable fragment isolated from APS, bound well to PD-1 and inhibited PD-L1-based T-cell depletion [138]. For example, a traditional Chinese medicine blend known as Bu Zhong Yi Qi Tang (BYD), which contains APS as a key ingredient, effectively reduced PD-L1 expression in gastric cancer tissues and targeted PD-1 and PD-L1 within tumors through the PI3K/Akt/mTOR pathway, resulting in significant suppression [136]. In addition, by inhibiting the Akt/mTOR/p70S6K pathway in tumor-bearing mice, APS downregulated the expression of PD-L1 on the surface of breast cancer 4T1 cells and colorectal cancer CT26 cells, and blocked PD-L1-induced T-cell exhaustion [137]. APS can also decrease PD-L1 expression in cisplatin-resistant melanoma cells, thus increasing their sensitivity to chemotherapy by acting on the PD-L1/PI3K/Akt pathway [66]. In a small retrospective study involving patients with lung cancer treated with ICI therapy, the addition of APS enhanced the antitumor effects of immunotherapy by reducing inflammation, lowering the prognostic marker NLR, and changing the relative proportion of circulating lymphocytes [139].

In combination with anticancer drugs, there was no substantial difference between the effects of ixabepilone and APS on tumor suppression in combination with PD-1 inhibitors, suggesting that APS prevented tumorigenesis or progression in the presence of increased T-cell activation, enhancing the synergistic effect with anti-PD-L1 [138].

APS, derived from the traditional Chinese herb *Astragalus membranaceus*, enhances anticancer immune responses by modulating the PI3K/Akt/mTOR and Akt/mTOR/p70S6K pathways, which are crucial for cell survival, proliferation, and metabolism. Suppressing these pathways diminishes cancer cell survival and PD-L1 expression on their surfaces. ICIs, which block PD-L1, aid the immune system in recognizing and eliminating cancer cells. Thus, APS not only acts independently but also potentially augments established cancer treatments by boosting the body’s natural immune response to tumors.

### 3.2. APS Enhances Relay Cellular Immunotherapy

Cellular immunotherapy, also known as permissive immune cell therapy, is an important antitumor therapy. This therapy is carried out by isolating autologous tumor-infiltrating lymphocytes (TILs) or peripheral blood lymphocytes (including lymphokine-activated killer (LAK) cells, CD3-activated killer (CD3-AK) cells, cytokine-induced killer (CIK) cells, and DC-CIK cells), which are activated or genetically modified in vitro, to amplify immune cells with antitumor activity (chimeric antigen receptor T (CAR-T) cells, T-cell receptor-engineered T (TCR-T) cells, and chimeric antigen receptor natural killer (CAR-NK) cells), equip them with the capability of tumor antigen-targeting recognition, and then infuse them back into the tumor patient’s body, thereby amplifying the cellular immune function in the patient’s body and improving the antitumor effect [140,141].

With regard to the effects on peripheral lymphocytes, when APS was combined with CIK cells, the antitumor response of DC-CIK cells was stronger than that of normal DC-CIK cells [142,143,144,145,146,147]. APS can promote the expression of specific markers on the surface of DC on the one hand and significantly enhance the DC-induced T-cell proliferative response; on the other hand, it can enhance the effect of CIK on CD3^+^ and CD4^+^ T lymphocytes and exert an antitumor effect. For instance, a study conducted by Zhang and Wang demonstrated that DC-CIK cells induced by APS not only exhibited increased proliferation, but also displayed enhanced cytotoxicity against A549 lung adenocarcinoma and Eca-109 esophageal cancer cells [142,143]. Another study by Zhang revealed that APS used in conjunction with CIK cell therapy effectively controlled tumor growth in the intermediate and advanced stages of NSCLC. This combination led to a significant increase in CD3^+^ and CD4^+^ T cells in the peripheral blood, improved the overall physical well-being of patients, and alleviated symptoms associated with qi deficiency syndrome, while maintaining a better safety profile [144]. In addition, the cytotoxicity of CIK cells combined with APS was higher than that of CIK cells alone in HeLa and SKOV3 cells. In vivo results showed that CIK combined with APS significantly inhibited the growth of HeLa-implanted tumors, with an inhibition rate of 80.6% [148].

Stilbene Silver San Liang San (SLS), a Chinese herbal formula consisting of *Astragalus*, honeysuckle, Angelica sinensis, Glycyrrhiza glabra, and centipede, is used as an adjuvant treatment for B-cell lymphomas with CD19 CAR-T cells. At the cellular level, SLS dose-dependently promoted the killing effect of unmodified T cells and anti-CD19 CAR-T cells on Raji cell lines, while at the same time reducing T-cell and CAR-T cell exhaustion to a certain extent, promoting the proliferation of anti-CD19 CAR-T cells, decreasing the levels of IL-6, IL-10, and TNF-α, and increasing the levels of granzyme B. SLS in in vivo studies effectively increased the anti-B-cell lymphoma function of anti-CD19 CART cells, prolonged the survival of mice, and decreased the levels of IL-6, GM-CSF, and IL-17. The specific mechanism of action of SLS is mediated by the IL-17 signaling pathway and the CD8^+^ T-cell apoptosis pathway, which regulates CAR-T cells [149]. In addition, a project to explore the role and mechanism of APS in promoting CAR-T cell therapy for HCC based on T-cell subpopulation differentiation is underway [150].

### 3.3. Utilizing APS as an Adjunct to Enhance the Effectiveness of Cancer Vaccines in Therapy

Cancer-preventive and therapeutic vaccines are key components of cancer-specific active immunotherapy. These vaccines stimulate a patient’s immune system against tumor-specific antigens [151]. However, many vaccines, especially those using peptides and nucleic acids, have a limited ability to activate the immune system. Therefore, their combination with potent adjuvants is crucial to achieve the desired therapeutic results. 

Studies have shown that APS can be used as an enhancer of DC-based cancer immunotherapy vaccination [55]. In an animal model of breast cancer, researchers evaluated the adjuvant effects of two polysaccharides extracted from the roots of *Astragalus* and Radix et Rhizoma Ginseng on DC cancer vaccine-treated mouse 4T1 cells and found that dextran extracted from APS could be an effective alternative to bacterial lipopolysaccharide (LPS) as an adjuvant for the formulation of DC-based cancer immunotherapy vaccines. This combination not only significantly increased the number of CD40, CD80, and CD86 markers in DCs, but also upregulated the secretion of IL-6, TNF-α, and IL-1β. In an animal model of anal sarcoma, the researchers used APS instead of TNF-α to induce maturation of bone marrow-derived DCs, which were then sensitized with the S180 tumor antigen, thereby obtaining a DC tumor vaccine. After immunotherapy in mice, antitumor cytokines such as TNF-α and IL-12 were produced to achieve anticancer effects [152]. It is also possible that the mechanism of this anticancer effect is related to the increase in thymic and splenic indices in mice, which promotes the transfer of the Th1/Th2 imbalance to cellular immunity, in which Th1 cells predominate [153]. In addition, a complex of *Astragalus* with an aluminum hydroxide adjuvant may enhance anti-breast-cancer activity by inducing an immune response to the basic fibroblast growth factor (bFGF) tumor vaccine in the TME, causing it to produce potent bFGF-specific cytotoxic T-lymphocyte activity [154]. In contrast to typical vaccine adjuvants, APS does not lead to a “cytokine storm” caused by the overexpression of vaccine adjuvants [28].

Although APS and similar herbal therapies have demonstrated encouraging outcomes in both preclinical and clinical trials, achieving complete cancer remission without side effects remains a significant challenge. Ongoing research into APS and other traditional herbal medicines is a vital path in devising more holistic and less detrimental cancer treatment methods. By blending traditional medicine with contemporary medical practices, we can offer a more thorough and patient-centered approach to cancer care.

## 4. Clinical Trials on the Antitumor Effects of APS

Although no antineoplastic drugs have been able to achieve complete remission of cancer without adverse effects, traditional Chinese herbs have been used for thousands of years in cancer treatment in China, Japan, and other East Asian countries, alleviating the symptoms in patients with cancer, and also mitigating the adverse effects and complications caused by chemotherapy and radiotherapy. These findings further strengthen the complementary role of herbs in modern cancer treatment and provide more options and possibilities for comprehensive treatment [155].

Among these herbs, *Astragalus* has attracted attention for its potential efficacy in cancer treatment. Several randomized clinical trials have demonstrated that *Astragalus*-based interventions can reduce cancer symptoms, improve the quality of life and immune function, increase plasma nerve growth factor levels, and slow the progression of chemotherapy-induced peripheral neuropathy [78,156]. Currently, *Astragalus* extracts or preparations [157] have been used in clinical trials for their antitumor effects on digestive, respiratory, epithelial tissue, and hematological malignancies (Table 3) [158,159,160]. 

### 4.1. Malignant Tumors of the Digestive System

According to the fifth edition of the Classification of Tumors of the Digestive System published by the World Health Organization (WHO) in 2019, malignant tumors of the digestive system include esophageal cancer, gastric cancer, cancer of ampullar carcinoma, gobelet cell adenocarcinoma, colorectal cancer, HCC, and pancreatic cancer [182]. APS has been used in clinical antitumor trials for gastric, colorectal, liver, and pancreatic cancers. 

Gastric cancer (GC) is the fourth most prevalent cancer globally [183]. Phytochemicals protect against GC through various mechanisms, including the inhibition of cell growth, the induction of cell death and autophagy, the prevention of cell invasion and migration, hindering angiogenesis, combating *Helicobacter pylori* infection, and impacting the TME [92,184]. Polysaccharides, especially those derived from fungi, algae, tea, APS, Dendrobium, and other edible plants and herbs, are recognized for their anticancer properties against GC [62]. The anti-GC effects of APS are manifested not only in the enhancement of immune activity and the direct inhibition of tumor cell proliferation but also in the alleviation of adverse reactions to anticancer drugs [57,68,170,185,186,187,188,189].

FOLFOX is a globally recognized first-line chemotherapeutic regimen for GC. However, 5-FU and oxaliplatin (L-OHP) may cause adverse reactions such as peripheral neurotoxicity, total hematopenia, and severe gastrointestinal responses [188,189]. Clinical trials have demonstrated that APS injections, when combined with FOLFOX, can reduce gastrointestinal reactions and leukopenia, offering new possibilities for patients with GC to enhance clinical outcomes and physical well-being while mitigating adverse effects [170]. Additionally, clinical studies have investigated the enhancement of advanced GC sensitivity to ICIs by modulating intestinal flora and exploring the metabolomics-based effects of Aiqi Shi Cao Tang (APS) in modulating the gastric environment for low-grade gastric intraepithelial neoplasia treatment [169,190].

Colorectal cancer (CRC) is the third and second most common cancer among men and women, respectively. The reasons for the increase in its occurrence are not fully understood, although genetics, lifestyle, obesity, and environmental factors may play roles [191]. Various studies have demonstrated that herbal medicines containing *Astragalus* in combination with chemotherapy increased the tumor remission rate, reduced chemotherapy-related side effects such as neutropenia, anemia, nausea, vomiting, and neurotoxicity, and enhanced the quality of life of patients with CRC compared to chemotherapy alone [175,176]. Two additional studies on APS combined with FOLFOX4 demonstrated similar effectiveness [177,178]. Shenqi Fuzheng Injection is a Chinese herbal formula consisting of *Astragalus* and Codonopsis. This injection was approved by the Chinese Food and Drug Administration in the 1990s [192]. Clinical studies have been conducted on the use of Shenqi Fuzheng Injection in combination with chemotherapy or radiotherapy for the treatment of lung, breast, and colorectal cancers, and some studies have suggested that it may play an important role in the treatment of advanced cancers by improving tumor response and reducing the toxicity of chemotherapy [193,194,195]. A meta-analysis showed that Shenqi Fuzheng Injection combined with chemotherapy regimens (FOLFOX regimen, XELOX regimen) in the treatment of CRC could improve the efficacy of chemotherapy and the quality of survival, enhance cellular immune function, and reduce adverse events such as leukopenia, thrombocytopenia, and gastrointestinal toxicity [194]. Another meta-analysis showed that the efficacy of ginseng *Astragalus* tonic injection added to conventional chemotherapy for the treatment of lung cancer, breast cancer, and gastrointestinal tract tumors was better than that of conventional chemotherapy alone, and the objective tumor responses were higher than those in the conventional chemotherapy group. Compared with the conventional chemotherapy group, the combination treatment with Shenqi Fuzheng Injection increased NK, CD3^+^, and CD4^+^ levels and the CD4^+^/CD8^+^ ratio [193]. AC591, a standardized extract of *Astragalus* and Gui Zhi Wu Tang, is a herbal formulation that improves numbness and pain in the limbs. AC591 can prevent oxaliplatin-induced neuropathy without reducing its antitumor activity against CRC. It is a promising adjunctive drug for the clinical relief of sensory symptoms [196]. An open clinical study of Chinese herbal medicine (*Astragalus*/tansy) in combination with a novel tumor-lysing vaccine for the treatment of recurrent or metastatic colorectal cancer is underway [174].

Despite the predominant use of Western medical interventions for HCC, the overall outlook for patients is frequently bleak, highlighting the critical need for preventive strategies against HCC [197]. A comprehensive analysis indicated that administering APS in conjunction with transcatheter hepatic artery chemoembolization for HCC therapy surpassed the performance of other Chinese herbal injections, enhancing clinical efficacy and treatment outcomes [198]. A recent clinical investigation was conducted on the combined use of *Astragalus*/tansychia and a novel lysosomal virus/fusion cell vaccine to treat advanced HCC and lung cancer [167].

The pathogenesis of pancreatic cancer remains unclear. The onset is inconspicuous, with symptoms often evident only in the middle or late stages. Although surgical removal is the ideal approach for patients with intermediate and advanced pancreatic cancer, chemotherapy in conjunction with traditional Chinese medicine can alleviate clinical symptoms, improve the TME, eliminate tumor recurrence factors, and reduce focal metastasis, thereby enhancing efficiency and minimizing toxicity [199].

During the investigation into APS and Shenqi Fuzheng Injection (composed mainly of Radix et Rhizome Ginseng and Radix Astragali) for treating gastrointestinal diseases, APS not only slowed pancreatic tumor growth but also reduced tumor-associated fibroblasts’ ability to secrete βig-H3, thereby promoting the expansion of CD8^+^ T cells and inhibiting the transition of macrophages to the M2 type, and consequently enhancing the pancreatic tumor’s sensitivity to gemcitabine and other therapies. An observational study found that combining APS with chemotherapy significantly prolonged the survival of patients with metastatic pancreatic cancer compared with chemotherapy alone [173]. Therefore, a randomized double-blind trial is planned to compare the efficacy of APS combined with gemcitabine versus gemcitabine alone as a neoadjuvant treatment for pancreatic cancer [172]. 

Esophageal cancer is a common tumor of the digestive tract. Because patients with esophageal cancer may have obvious cellular immunosuppression after surgery, some relevant postoperative studies have been conducted. In a postoperative study of 60 patients with thoracic segment esophageal cancer, postoperative intravenous Shenqi Fuzheng Injection significantly improved the cellular immune function after modern two-field resection compared to the group without immunosupportive therapy [200]. Another study showed that four weeks of perioperative treatment with *Astragalus*-enhancing flour in 37 patients with esophageal cancer resulted in a more pronounced number of mesenchymal mast cells, focal lymphocyte aggregates, increased mesenchymal microvascular damage, and improved immune function [201]. A randomized open clinical trial is currently underway to evaluate APS in conjunction with concurrent chemoradiotherapy for the treatment of locally advanced esophageal cancer [180].

### 4.2. Malignant Tumors of the Respiratory System

Malignant tumors of the respiratory system include lung, nasal, and laryngeal cancers. APS is being used in clinical antitumor trials related to NSCLC and nasopharyngeal cancer.

NSCLC accounts for the majority of lung cancer cases, with a high rate of recurrence and limited efficacy of current treatments [202]. *Astragalus*-containing herbs are effective in preventing radiotherapy pneumonia as an adjunctive therapy during conservative radiotherapy [203]. Zhou et al. observed decreased cell growth and increased apoptosis in the H460 NSCLC cell line following administration of APS [204]. In addition, Gan et al. explored the potential of APS for clinical use in lung cancer. Combining platinum-based chemotherapy with *Astragalus*-based herbal treatments has shown enhanced efficacy in the treatment of advanced NSCLC [205]. By injecting APS in combination with the chemotherapeutic agents vincristine and cisplatin in patients with advanced NSCLC, it was found that APS significantly improved immune function and reduced the toxicity and side effects of chemotherapy and radiotherapy, leading to a better quality of life compared to chemotherapy alone. In light of these benefits, researchers have conducted clinical trials using APS to alleviate cancer-related fatigue induced by chemotherapy in patients with lung cancer, as well as for maintenance therapy after initial or subsequent chemotherapy and targeted therapy [162]. Studies have shown that Shenqi Fuzheng Injection can improve the clinical efficacy and reduce the radiotoxicity in patients with NSCLC [195]. Ten-perfect tonic soup, composed of *Astragalus*, ginseng, angelica, and other herbs, is commonly used for outpatient chemotherapy in patients with lung cancer. A quality-of-life questionnaire on anticancer drug therapy and another randomized, double-blind, placebo-controlled trial conducted on patients with NSCLC showed significant improvements in both physical symptoms and anorexia after treatment with decapod tonic soup [206,207]. Decapentaplegic tonic soup may also increase the regulatory activity of T cells by decreasing the Foxp3^+^ Treg population in patients with advanced pancreatic cancer, leading to immune enhancement with various combination therapies [208]. Aidi injection, a commonly used adjuvant chemotherapeutic agent in China, consists of plant extracts of *Astragalus*, *Echinococcus* sp., and ginseng, with antitumor, immunomodulatory, and attenuating acute or subacute chemotherapy toxicity. Aidi injection combined with radiotherapy can reduce myelosuppression, radiation pneumonitis, and radiation esophagitis after radiotherapy, and significantly improve the clinical efficacy and quality of life of patients with lung cancer [209,210].

Nasopharyngeal cancer is highly aggressive [211]. Although platinum-based chemotherapy is an effective treatment, the associated adverse effects limit its use [212]. A case–control study linked the use of APS to a decreased incidence of nasopharyngeal cancer [213]. 

### 4.3. Other Malignant Tumors

Malignancies of epithelial origin include squamous cell carcinoma, adenocarcinoma, colorectal carcinoma, and breast carcinoma. APS has been tested in clinical antitumor trials related to malignancies of epithelial tissue origin and hematological malignancies.

Metastasis accounts for 90 percent of breast-cancer-related deaths. Treatment options for metastatic breast cancer are limited, emphasizing the urgent need for innovative therapeutic strategies to address this critical issue [214]. Investigations have elucidated the mechanism of action of APS against breast cancer by examining the essential target proteins and mRNA expression. Seventy-three downregulated and seven upregulated genes were identified, with *EGFR* and *ANXA1* identified as the pivotal genes associated with breast cancer progression. Treatment with APS results in a significant decrease in EGFR levels and a marked increase in ANXA1 levels in breast cancer cells [215]. Furthermore, Wang et al. successfully developed a novel nanoparticle (quercetin-3’3-dithiobispropionic acid-Astragaloside-folic acid) that not only hindered drug release into the environment but also facilitated delivery into cancer cells, effectively inhibiting estrogen receptor α-positive breast tumors [216]. Studies have shown that APS treatment can reduce chemotherapy-induced toxicity, leading to research on stage II/III breast cancer patients receiving APS as adjuvant chemotherapy [168], underscoring the importance of ongoing clinical trials on APS in breast cancer care.

Concurrent chemoradiotherapy (CCRT) is a crucial treatment for patients with advanced head and neck squamous cell carcinoma (HNSCC) [217]. A phase II double-blind, randomized trial compared CCRT with APS injection to CCRT with placebo for advanced pharynx or larynx squamous cell carcinoma, and found that the CCRT and APS group had improved safety, enhanced quality of life, and reduced adverse events commonly associated with intense anticancer therapies in patients with HNSCC undergoing CCRT [218].

The enhanced antitumor effects of *Astragalus* injection in combination with chemotherapy may improve the short-term prognosis and clinical regression of acute lymphoblastic leukemia (ALL) in children [219]. Remission induction chemotherapy can reduce myelosuppression induced by chemotherapeutic agents, increase neutrophil counts, and reduce the incidence and duration of infections in children with ALL [220].

## 5. Comparison of Anticancer Effects of *Astragalus* Polysaccharide and Other Polysaccharides

In recent years, polysaccharides isolated from natural sources such as plants, animals, fungi, and seaweeds have received increasing attention owing to their various pharmacological activities, including antitumor, immunomodulatory, antioxidant, and anti-inflammatory activities [221,222,223,224]. Pectin polysaccharides from plants, β-glucans from mushrooms, and sulfated polysaccharides from seaweeds have been shown to possess antioxidant and immunomodulatory activities [225,226,227]. *Astragalus*, mushrooms, *Ganoderma lucidum*, and fucoidan polysaccharides are used in domestic and international markets.

### 5.1. Astragalus Polysaccharide and β-Glucan

β-glucan is found in the cell walls of mushrooms, yeast, and grains such as oats and barley [228]. Both β-glucan and *Astragalus* polysaccharide are known for their effects on modulating immune function. However, β-glucan generally activates white blood cells such as macrophages and neutrophils, whereas *Astragalus* polysaccharide tends to enhance the body’s production of various immune cells [229]. In terms of applications, β-glucans are commonly used in immune-enhancing supplements and food additives, while *Astragalus* polysaccharide is often used in herbal formulations to enhance vitality and treat diseases such as colds or flu [230,231]. Among the β-glucans, shiitake polysaccharide is a typical T-cell activator that enhances the immune function of immune cells such as T cells, macrophages, and natural killer cells, and has antitumor effects [232]. Polysaccharide peptides extracted from medicinal mushrooms induced Molt4 leukemia cells to enter the S phase, eventually leading to apoptosis [233,234]. *Ganoderma lucidum* polysaccharides reduced ERK expression and inhibited the proliferation of MT-1 human breast malignant carcinoma cells tumor cells through the ERK signaling pathway [235]. *Lycium barbarum* polysaccharides (LBPs) enhanced macrophages in a dose-dependent manner [236]. They exhibited antitumor activity against many types of cancer cells, and inhibited tumor growth in nude mice by inducing apoptosis and cell cycle blockade [237,238]. Ke et al. reported that LBP inhibited the growth of the human bladder cancer cell line BIU87 and induced apoptosis in BIU87 cells [239].

### 5.2. Astragalus Polysaccharide and Pectin Polysaccharide

Pectin is a polysaccharide derived from citrus fruits and apples that can form gels and is more structurally complex with more branching than APS [240,241]. Pectin is primarily used to improve digestive health and reduce cholesterol [242]. In addition, APS has antioxidant properties that help neutralize free radicals and protect cells from oxidative damage [243]. Pectin mainly produces reactive oxygen species to induce apoptosis, which is the opposite mechanism of action of APS [244]. Pectin is widely used as a gelling agent in jams and preserves by the food industry, whereas APS is mainly found in dietary supplements and traditional medicine [23,245].

In addition to their benefits as soluble dietary fibers, modified pectins, particularly modified citrus pectins, have beneficial effects on the development and spread of malignant tumors, including colon and breast cancer [245]. Pectin-derived chemicals slow cell development and promote apoptosis [246]. The combination of the multiplant compounds BreastDefend or ProstaCaid with modified citrus pectin reduced the invasive potential of highly metastatic human breast cancer MDA-MB-231 cells and may prevent the development of breast cancer in mice by reducing angiogenesis [247]. Furthermore, in addition to activating macrophages, citrus pectin inhibited activator protein 1 and NF-κB signaling and activated LPS/TLR-4 signaling [248]. Citrus and apple pectin inhibited the growth of MDA-MB-231, MCF-7, and T47D breast cancer cells by inhibiting the S phase and G1 phase or G2/M phase of the cell cycle. They stimulated the intrinsic mitochondrial apoptotic pathway by generating reactive oxygen species and inhibiting matrix metalloproteinases in breast cancer cells, thereby inducing cell death through DNA damage [249].

### 5.3. Astragalus Polysaccharide and Xanthanate Gum

Xanthan gum is a naturally occurring polysaccharide obtained from *Xanthomonas campestris* with a β-d-glucose linkage in its main chain, similar to cellulose [250]. Unlike xanthan polysaccharides, which have therapeutic properties, xanthan gum is mainly used in foods as a thickening and stabilizing agent and has few direct health benefits [251]. However, some studies have confirmed that xanthan gum has antitumor effects, in addition to acting as an adjuvant to modulate immunity [252]. In vitro cultured macrophages were induced to produce IL-12 and TNF-α in an MyD88-dependent manner by xanthan gum, being recognized by TLR4. In in vivo experiments, oral administration of xanthan gum significantly delayed tumor growth and prolonged survival in mice with melanoma and enhanced NK cell activity and CD8^+^ T-cell tumor specificity [253].

### 5.4. Astragalus Polysaccharide and Marine Complex Polysaccharide Substances

In the search for novel polysaccharides with antitumor properties, marine resources have attracted considerable attention because of their unique biological activities [254,255,256]. Fucoidans are biologically active sulfated polysaccharides found in brown algae. They have great structural diversity and a wide range of bioactivities and have been shown to be useful in antitumor therapy, cardiovascular disease risk reduction, and the treatment of gastric ulcers, which differ from the therapeutic effects of APS for a large part of the population [257]. Studies have shown that fucoidan exerts anticancer effects by mediating different signaling pathways to regulate apoptosis, inhibit tumor metastasis, and enhance the toxic effects of chemical drugs [258,259,260,261]. The anticancer properties of fucoidans are believed to be determined by their chemical structures and are closely related to their molecular weights; the larger the molecular weight and the higher the water solubility of polysaccharides, the stronger their antitumor activity [260]. As an adjuvant, fucoidan enhances the efficacy of ICIs in the treatment of melanoma and metastatic lung cancer [260,262,263]. In a clinical study, the combination of fucoidan and tamoxifen showed favorable efficacy for the treatment of breast cancer, and no adverse effects were detected [264]. This suggests that fucoidan could be used as an effective adjuvant in combination with other hormonal therapeutic agents. In addition, sulfated polysaccharides purified from marine microalgae showed cytotoxicity against the colorectal cancer HTC-116, breast cancer MCF-7, and human leukemia HL-60 cell lines [265].

### 5.5. Production of Astragalus Polysaccharide

Although APS has shown promise in laboratory and clinical trials, research on its specific mechanisms and long-term effects is relatively limited. APS is less well studied than longer-established and more widely used polysaccharides such as β-glucan and pectin. In terms of accessibility and assessment of production costs, APS is primarily derived from *Astragalus* root, which may limit the global availability and scale of production. Fucoidan may also be affected by the quality and safety of brown algae due to environmental factors such as marine pollution. The cost is high compared to that of more readily available sources, such as citrus fruits and marine complex polysaccharides. While APS is generally considered safe, there may be safety concerns and side effects when used at high doses or over long periods of time. For example, when associated with certain drugs for autoimmune diseases, immune activation may result. APS requires more careful regulation than food-grade polysaccharides such as gum and xanthan gum. Thus, while APS offers significant advantages in terms of immune enhancement and anti-inflammatory and antioxidant properties, it may have disadvantages in terms of the depth of research, cost-effectiveness, and safety. When selecting a suitable polysaccharide supplement, consumers should consider these factors and make decisions based on their health status and needs.

## 6. Conclusions and Outlook

In the current landscape of cancer therapy, APS has been recognized for its immunomodulatory potential as a beneficial bioactive substance with various promising applications. APS is considered a suitable adjuvant drug with no significant safety issues [266]; however, numerous studies have highlighted its role in enhancing patient quality of life by reducing cancer-related fatigue and symptoms and modulating inflammatory responses [23,52].

This review comprehensively evaluates the anticancer mechanisms of APS, focusing on its potential use in cancer immunotherapy. The results underscore the ability of APS to activate and modulate immune cells, disrupt the cell cycle, induce cancer cell apoptosis, trigger cellular autophagy, and regulate signal transduction pathways, especially showing potential in enhancing the sensitivity of antitumor therapy, attenuating the treatment-related side effects, and reversing drug resistance. The advantages and disadvantages of APS in comparison with other polysaccharides were also considered.

Nonetheless, several key issues must be addressed to realize the widespread clinical application of APS. First, the bioavailability of APS is low, mainly due to the complex structure of its macromolecules, making its absorption in the gastrointestinal tract inefficient [267]. In addition, owing to differences in plant sources and extraction methods, there may be variability in the APS composition, which affects the consistency and reproducibility of the clinical results [268]. Therefore, it is crucial to develop standardized extraction and purification processes to ensure product quality. Second, although APS is generally considered safe, the issue of dose optimization in terms of potential interaction with other drugs and side effects in specific populations requires further research [269].

Future studies should focus on gaining a deeper understanding of the specific mechanism of action and pharmacokinetic properties of APS and revealing the molecular targets of APS through high-throughput screening, genomic studies, and advanced imaging techniques to develop effective clinical strategies. Meanwhile, exploring innovative delivery methods, such as extended-release gel systems for injection, may provide solutions to improve its absorption and sustained release in vivo and reduce the need for frequent dosing in long-term therapy [270]. Through these efforts, APS is expected to play a greater role in modern healthcare systems, especially in the field of cancer therapy.

## Figures and Tables

**Figure 1 pharmaceuticals-17-00636-f001:**
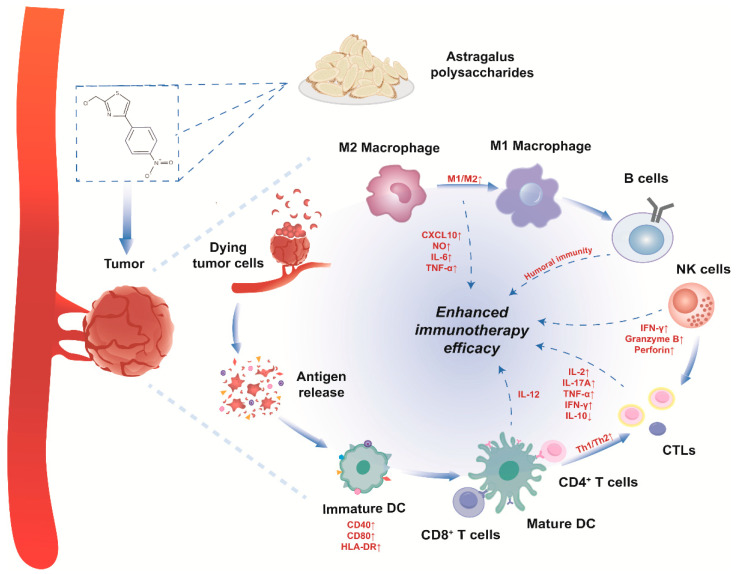
The impact of APS on immune cell activity in the tumor microenvironment. APS promotes the expression of CD40, CD80, and HLA-DR on dendritic cells, facilitating their maturation process. These mature dendritic cells subsequently stimulate the production of IL-12, direct antigens towards lymphocytes, shift Th2 cell populations towards Th1, and reinforce the immune response of T cells and NK cells. This leads to an increase in IL-2, TNF-α, IFN-γ, and IL-17A by T cells, accompanied by a reduction in IL-10. NK cells exhibit elevated levels of IFN-γ, granzyme B, and perforin; APS also enhances the proportion of M1/M2 macrophages and promotes the secretion of NO, CXCL10, IL-6, and TNF-α, thereby enhancing the humoral immunity of B cells. The solid line is the effect that one cell can have on another cell, and the dashed line is the effect that the cell has on immunity. Down and up directions means more or less related factors.

**Figure 2 pharmaceuticals-17-00636-f002:**
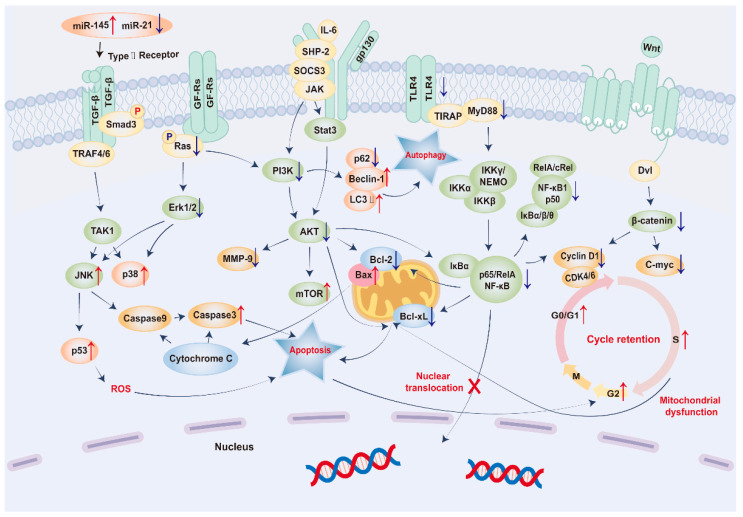
Diagram of the mechanism by which APS induces apoptosis in cancer cells by interfering with the cell cycle, inhibiting signal transduction pathways, and inducing apoptosis. This involves the induction of S, G2/M, and G0/G1 cell cycle arrest via the Wnt-β-catenin pathway. It is involved in the inhibition of PI3K/Akt, ERK/MAPK, and NF-κB pathways in tumor therapy and induces apoptosis in cancer cells by inhibiting the expression of Bcl-2 proteins and miRNAs and upregulating the expression of p53 proteins. Yellow represents the factor on the receptor, which is also the initiating factor of the pathway; Green represents factors on the conventional signaling pathway; Orange represents factors involved in both apoptosis and autophagy; Pink like Bax represents both apoptosis and cell cycle involvement; Earthy yellow represents cell cycle related factors; Blue represents cytokines involved in apoptosis alone. The red up arrow indicates that *Astragalus* polysaccharide leads to the increase of related factors, and the blue down arrow indicates that *Astragalus* polysaccharide leads to the decrease of related factors.

**Figure 3 pharmaceuticals-17-00636-f003:**
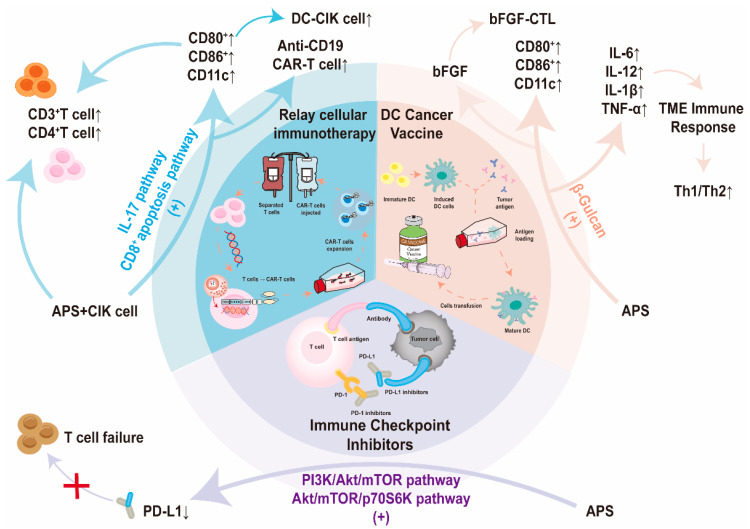
Effects of APS on ICIs, CAR-T therapy, and DC cancer vaccines within the TME. This involves the enhancement of ICIs through the PI3K/Akt/mTOR pathway. APS-induced activation of DCs improves the effectiveness of adoptive immunotherapy with cytokine-induced killer (CIK) cells and APS components, particularly abundant in dextran, which act as potent adjuvants for DC-based cancer immunotherapy vaccines. The blue arrow belongs to the APS-stimulated adoptive cell immunotherapy arrow, the purple arrow belongs to the APS-stimulated immune checkpoint inhibitor arrow, and the pink arrow belongs to the APS-stimulated DC cancer vaccine arrow. The black up arrow indicates an increase in a certain factor during APS stimulation, and the black down arrow indicates a decrease in a certain factor during APS stimulation.

**Table 1 pharmaceuticals-17-00636-t001:** Effects of APS on immune cells (DC, B, T, macrophage, NK).

Model Source	Dosage	Effect of APS on Immune Cells	Reference
C57BL/6 mice	200 μg/mL, 24 h	bone marrow-derived dendritic cells(BMDCs) ↑ Splenic DCs in mice 6 h ↑18 h ↓ Human peripheral blood DCs (PBDCs) 6 h ↑	[26]
C57BL/6J mice	45 μg/mL, 24 h	BMDC ↑	[27]
BALB/c mice	200 μg/mL, 24 h	CD8^+^ T ↑ CD4^+^ T ↑ pDC ↑	[28]
BALB/c mice	50 μg/mL, 24 h	BMDC ↑ (CD4^+^ T/Treg cell)/(CD8^+^ T/Treg cell) ↑ CD4^+^ T ↑	[29]
C57BL/6 mice	16 mg/mL, 48 h	M1/M2 ↑ CD11c(high) CD45RB(low) DCs ↑ Th1/Th2 ↑	[30]
C57BL/6 mice	50 mg/kg, 18 h	CD11c^+^ DC ↑ CD3-NK1.1^+^ ↑ CD8^+^ T ↑	[31]
Kunming mice C57 mice	100 mL/kg, 3 days	CD4^+^ T/CD8^+^ T ↑	[32]
C57BL/6J mice	0.2 mg/mL	Treg ↓ BMDC ↑ CD3^+^CD8^+^ T ↑ MHC-I^+^CD11c^+^ cells ↑	[33]
Swiss-Hauschka (ICR) mice	5 mg/mL, 14 days	Th1/Th2 ↑ CD4^+^ T/CD8^+^ T ↑	[34]
Spleen lymphocyte	0.156 mg/mL	Th1/Th2 ↑ CD4^+^ T/CD8^+^ T ↑	[35]
BALB/c mice	300 μg/mL	Macrophage (M2 → M1) ↑	[36]
Mouse peritoneal macrophages	100 ng/mL, 24 h	Raw 264.7 ↑	[37]
Murine macrophage RAW 264.7 cells	2000 μg/mL, 24 h	Raw 264.7 ↑ (IL-1α, IL-1β, IL-6 ↑, NO ↓)	[38]
The fresh lung tissues of adult pigs	500 μg/mL or 1000 μg/mL, 24 h	Raw 264.7 ↑ (TNF-α, NO ↑)	[39]
BALB/c mice or C3H/HeJ mice	100 μg/mL, 6 h	Raw 264.7 ↑, B ↑ (IL-1β, TNF-α ↑)	[40]
ICR mice	10 g/kg	CD8^+^ T ↑ CD4^+^ T ↑ NK ↑ CTL ↑	[41]
BALB/c mice	100, 200, 300 mg/kg, 30 days	Raw 264.7 ↑ NK ↑ B ↑ T ↑	[42]
BALB/c mice	100 mg/kg	NK ↑	[43]

Arrows in down and up directions means more or less related factors: ↑ increased, ↑ decreased.

**Table 2 pharmaceuticals-17-00636-t002:** Effect of APS on cell cycle, signal transduction pathways, and apoptosis.

Model Source	Dosage	Effect of APS Cell Cycle, Signal Transduction Pathways, and Apoptosis	Reference
Kunming mice	150, 300 mg/kg, 15 days	G1 phase retention ↓ S phase retention ↑	[56]
MGC-803 cell	200, 400, 800 μg/mL, 24 h	G0/G1 phase retention ↓ S phase retention ↑ G2/M phase retention ↓	[57]
HepG2 cell	200, 400, 800 μg/mL, 72 h	S phase retention ↑	[58]
4T1 cell	50, 100, 200, 500, 1000 μg/mL, 72 h	G2/M phase retention ↑	[59]
4T1 cell	50 μg/mL, 24 h	G1 phase retention ↓ G2/M phase retention ↑	[60]
MCF-7 cell	500, 1000 μg/mL, 24 h	G1 phase retention ↑	[61]
SCG-1 cells	25, 50, 100 mg/L, 12 h	G0 phase retention ↑	[62]
CNE-1 cell	200 μg/mL, 24 h	G0/G1 phase retention ↑ S phase retention ↑	[63]
AGS cell	200 μg/mL, 24 h	PI3K/Akt pathway ↓	[64]
B16F10 cell	1–5 mg/mL, 24 h	PI3K/Akt pathway ↓	[65]
C57BL/6 mice	200 mg/kg, 24 h	PI3K/Akt pathway ↓	[66]
RAW264.7 cell	0–100 μg/mL, 24 h	NF-κB/MAPK pathway ↑ P65 protein ↑ G2/M phase retention ↑	[67]
GC-SGC-7901 cell GCSGC-7901/ADR cell GES-1 cell	100–400 μg/mL, 24 h	p-AMPK level ↑	[68]
A549 cell NCI-H358 cell	20, 40 mg/mL	NF-κB pathway ↓	[69]
PANC-1 cell	0, 1, 5, 10, 15, 20 mg/mL	NF-κB P65 ↓ TLR4/NF-κB pathway ↓	[70]
A549 cell BMSC cell	50 μg/mL	RAS, ERK, NF-κB p65 protein ↓ TP53, caspase-3 protein ↑	[71]
HepG2 cell Diethylnitrosamine-induced HCC in rats	60, 120, 240 mg/kg (rats) 20, 40, 80 µg/mL (HepG2 cell)	TGF-β/MAPK/Smad pathway ↑ miR-145 ↑, miR-21 ↓ pSmad3L→pSmad3C ↑	[72]
HepG2 cell	40 mg/mL	TGF-β/Smad pathway ↑ pSmad3L→pSmad3C ↑	[73]
H22 cell Kunming mice	100, 200 mg/kg, 16 days	Bax ↑ Bcl-2 ↓	[74]
MDA-MB-231 cell BALB/C mice	200, 400 mg/kg, 21 days	Bax, Caspase7, Caspase9 ↑ Bcl-2 ↓	[75]
HepG2 cell	100, 200 mg/L, 48 h	Bcl-2 ↓ Caspase3 ↑ β-catenin, c-myc, Cyclin D1 mRNA ↓	[76]
SKOV3 cell	800 µg/mL, 24 h	Bcl-2 ↓ Bax, Caspase3 ↑	[77]
BALB/C mice CNE-1 cell CNE-2 cell SUNE-1 cell	40 µg/mL, 48 h	Bcl-2 ↓ Bax, Caspase3, Caspase9 ↑	[78]
OV-90 cell SKOV-3 cell	1 mg/mL, 24 h	miR-27a ↑ Caspase3 ↑	[79]
OS MG63 cell	10 mg/mL, 24 h	miR-133a ↑ S phase retention ↑ CyclinD1 ↓ p21 ↑ Bcl-2 ↓ Bax, Caspase3, Caspase9 ↑	[80]
A549 cell NCIH1299 cell	0, 5, 10, 20 µg/mL,	miR-195-5p ↑	[81]
H460 cell	0–30 mg/mL, 24/48 h	P53, P21, P16 ↑ Notch1,Notch3 ↑ Bcl-2 ↓ Bax, Caspase8 ↑	[82]
CD133^+^/CD44^+^ cell	0, 12.5, 25, 50 mg/mL, 48 h	Fas Caspase 3, Caspase 9, Fas, Bax ↑ Class III PI3K, Beclin 1 ↑ Bcl-2, XIAP ↓ LC3-I ↓LC3-II ↑	[83]
Hep3B cell BALB/c mice	10 mg/L	CHOP ↑ Bcl-2 ↓ Bax, Bim, Caspase3 ↑ OGT ↓OGA ↑ O-GlcNAc ↓	[84]

Arrows in down and up directions means more or less related factors: ↑ increased, ↑ decreased.

**Table 3 pharmaceuticals-17-00636-t003:** Clinical investigations of APS in cancer.

Number	Pharmaceutical Ingredient	Disease	Status	Phase
NCT01802021 [161]	*Astragalus*-based Formula	Non-small-cell lung cancer	Recruiting	II/III
ChiCTR2300068199 [162]	*Astragalus* polysaccharides, Paclitaxel; Carboplatin	Non-small-cell lung cancer	Not Recruiting	−
[163]	*Astragalus* polysaccharides, Iodide 125	Lung cancer	Not Recruiting	II/III
[25]	*Astragalus* polysaccharides, Vinorelbine; Cisplatin	Non-small-cell lung cancer	Not Recruiting	II/III
[164]	*Astragalus* polysaccharides, Gemcitabine; Cisplatin	Non-small-cell lung cancer	Not Recruiting	II/III
ITMCTR2100004716 [165]	*Astragalus* polysaccharides, Apatinib	Extensive-stage small-cell lung cancer	Not Recruiting	−
ChiCTR2000040911 [166]	*Astragalus* polysaccharides, Carrelizumab; Apatinib	Lung cancer	Recruiting	−
ITMCTR2000003215 [167]	*Astragalus* polysaccharides, Tansychia	Hepatocellular carcinoma; Lung cancer	Recruiting	−
NCT03314805 [168]	*Astragalus* polysaccharides	Breast cancer	Not recruiting	II
ChiCTR2300076131 [169]	Chinese herbal decoction (contains *Astragalus* polysaccharides)	Low-grade gastric intraepithelial neoplasia	Recruiting	−
[170]	*Astragalus* polysaccharides, FOLFOX	Gastric cancer	Not recruiting	−
ChiCTR2300068896 [171]	*Astragalus* polysaccharides	Gastric cancer	Not recruiting	IV
NCT06234072 [172]	*Astragalus* polysaccharides, Gemcitabine	Pancreatic cancer	Not recruiting	II
[173]	Traditional Korean Medicine (contains *Astragalus* polysaccharides), CTX	Metastatic pancreatic cancer	Not recruiting	−
ChiCTR2000037982 [174]	*Astragalus*, Panax-notoginseng, Oncolytic vaccine	Colorectal cancer	Recruiting	−
[175,176]	*Astragalus* polysaccharides	Colorectal cancer	Not recruiting	−
[177]	FOLFOX 4, Ginseng, *Astragalus* polysaccharides, Atractylodes rhizome, Poria cocos, Coix seed, Sophora flavescens	Advanced colorectal cancer	Not recruiting	−
[178]	*Astragalus* polysaccharides, Oxaliplatin, Poria cocos, Atractylodes macrocephala Koidz, Pilosula	Colorectal cancer	Not recruiting	−
NCT01720563 [179]	APS, Cisplatin, Leucovorin, Tegafur plus uracil	Advanced pharyngeal/laryngeal squamous cell carcinoma	Not recruiting	II
NCT03611712 [180]	*Astragalus* polysaccharides	Locally advanced esophageal cancer	Not recruiting	II
NCT01720550 [181]	*Astragalus* polysaccharides	Advanced cancer	Not recruiting	IV

## Data Availability

No new data were created or analyzed in this study. Data sharing is not applicable to this article.

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
