# Peer review of "Anticancer Mechanism of *Astragalus* Polysaccharide and Its Application in Cancer Immunotherapy"

_pharmaceuticals, 2024, doi:10.3390/ph17050636_

Round 1

Reviewer 1 Report

Comments and Suggestions for Authors

This review summarizes the mechanism and application of  astragalus polysaccharide (APS) in cancer treatment. The study of APS in cancer treatment has always been one of the hot spots in traditional Chinese medicine. Therefore, this review has guidance and clinical significance. However, there are still some drawbacks.

1.     In the past five years, a few reviews about astragalus polysaccharide on  cancer treatment have been published, such as 1) “Li, Chun-xiao, et al. "Astragalus polysaccharide: a review of its immunomodulatory effect." Archives of Pharmacal Research 45.6 (2022): 367-389”; 2) “Yang, Qian, et al. "Advances in research on the anti-tumor mechanism of Astragalus polysaccharides." Frontiers in Oncology 14 (2024): 1334915” ; 3) “BEREZUTSKY, MIKHAIL ALEXANDROVICH, et al. "Anti-Cancer Activity of Astragalus Membranaceus--A Review." International Journal of Pharmaceutical Research (09752366) 13.3 (2021)”. Please provide your innovative contributions.

2.     The structure of this manuscript is not clear. For example, 1) Part 2, Anti-cancer mechanism of  Astragalus polysaccharide, summarizes different anticancer mechanisms,  but only its application in immunotherapy is summarized and discussed below; 2) Part 3 “APS-assisted cancer immunotherapy” and part 4 “Involvement of astragalus polysaccharides in tumor immunotherapy” are duplicates. Authors should remake the structure.

3.     Minor mistakes. For example, 1) In the Abstract, abbreviation should be accompanied by  full names; 2) In line 259, the title is inconsistent with the content.

Comments on the Quality of English Language

Minor editing of English language required

Author Response

请参阅附件。

Reviewer 2 Report

Comments and Suggestions for Authors

Reviewing the manuscript titled “Anti-cancer mechanism based on Astragalus polysaccharide and its application in cancer immunotherapy" the authors focus on astragalus polysaccharides (APS) as potential anti-cancer factors. Astragalus polysaccharides (APS) are key component in traditional Chinese medicine and cancer immunotherapy. They have anti-tumor properties and are safe. APS influences immune cells and promotes cell death, triggers autophagy, and impacts the tumor microenvironment. When combined with other therapies, APS can enhance treatment outcomes and reduce toxicity. In my opinion, the article is quite important because it has included several studies, which support that astragalus polysaccharides have become increasingly important for improving anti-tumor therapy's sensitivity, lowering its adverse effects, reversing anti-tumor medications' drug resistance, etc.

In my opinion it could be published if some critical/major issues are addressed:

-In order to make the mechanism of action of APS more understandable, a table should be introduced for each module, where the effect of APS on the signaling pathways is mentioned.

-In the immunotherapy section, more details need to be introduced on the effect of APS on immunomodulation, because we know it has a direct effect.

-The conclusions of the manuscript are not clear enough; they need a little reconstruction giving the basic issues of the use of APS as therapeutic agents.

Comments on the Quality of English Language

Minor editing of the English language is required.

Author Response

请参阅附件。

Reviewer 3 Report

Comments and Suggestions for Authors

This review by He and collaborators deals with the interest of Astragalus membranaceus polysaccharide (APS) in cancer treatment. This is an interesting topic as APS possess high medicinal value due to its multi-levels anti-cancer mechanisms. The review is well written and logically organized. It is a long manuscript, which contains a substantial amount of information and references. The bibliography analysis is sound and rather exhaustive. I believe this review can be improved pending a few modifications:

-        Abstract: please define the acronyms (“PICTs”)

-        Figures have to be enlarged

-        Please add a figure summarizing paragraphs 2.2 to 2.4 to illustrate the anti-tumor mechanisms mediated by APS

-        Page 6, line 259: please correct the title of paragraph 2.5

-        Paragraph 3.3: please precise the potential interest of APS in CAR-T therapy and DC cancer vaccine, with relevant references

-        Figure 2: the figure is not specific enough, and merely consists in a basic illustration of CAR-T, ICP and DC vaccines principles, with no clear illustration of APS impact. Please modify

Author Response

请参阅附件。

Reviewer 4 Report

Comments and Suggestions for Authors

The following points are my concerns/suggestions to the manuscript. The authors are requested to address it in the manuscript.

1. All the botanical names should be in italics. kindly revise it across the manuscript

2. APS (in abstract) should be abbreviated when it is said first.

3.  The image of the plant should be kept in the introduction part. 

4. small paragraphs should be avoided.  

5. authors perspective is a must rather having a conclusion

6. There is no sign of objective in the abstract. The review must contain an objective.

7. The period of study should be indicated in the abstract say for example: the review covers a decade of literature or two decades.. this will give the reader an overview of the script.

8.  A comparison between ASP with other natural polysaccharides should give strength to the manuscript. 

Comments on the Quality of English Language

Spell check and typo should be corrected

Author Response

请参阅附件。

Round 2

Reviewer 1 Report

Comments and Suggestions for Authors

Thanks for author's response for comment #2 and #3. However, authors didn't reply comment #1. 

1.     In the past five years, a few reviews about astragalus polysaccharide on  cancer treatment have been published, such as 1) “Li, Chun-xiao, et al. "Astragalus polysaccharide: a review of its immunomodulatory effect." Archives of Pharmacal Research 45.6 (2022): 367-389”; 2) “Yang, Qian, et al. "Advances in research on the anti-tumor mechanism of Astragalus polysaccharides." Frontiers in Oncology 14 (2024): 1334915” ; 3) “BEREZUTSKY, MIKHAIL ALEXANDROVICH, et al. "Anti-Cancer Activity of Astragalus Membranaceus--A Review." International Journal of Pharmaceutical Research (09752366) 13.3 (2021)”. Please provide your innovative contributions.

Reviewer 2 Report

Comments and Suggestions for Authors

The revised version of the manuscript finds me in complete agreement. The authors clearly answered all the issues I raised. I unreservedly recommend the publication of the work.

Reviewer 3 Report

Comments and Suggestions for Authors

I think the authors have adequately addressed my comments in the revised version of the manuscript, which has been significantly improved over the initial submission. I just believe the legend of Figure 2 could be slightly improved style-wise before publication.

Comments on the Quality of English Language

English is fine, but the legend of Figure 2 could be slightly improved style-wise before publication.
